# A Novel Piezoresistive MEMS Pressure Sensors Based on Temporary Bonding Technology

**DOI:** 10.3390/s20020337

**Published:** 2020-01-07

**Authors:** Peishuai Song, Chaowei Si, Mingliang Zhang, Yongmei Zhao, Yurong He, Wen Liu, Xiaodong Wang

**Affiliations:** 1Engineering Research Center for Semiconductor Integrated Technology, Institute of Semiconductors, Chinese Academy of Sciences, Beijing 100083, China; pssong@semi.ac.cn (P.S.); hyr617@semi.ac.cn (Y.H.); liuwen519@semi.ac.cn (W.L.); 2The School of Microelectronics & Center of Materials Science and Optoelectronics Engineering, University of Chinese Academy of Sciences, Beijing 100049, China; 3Beijing Academy of Quantum Information Science, Beijing 100193, China; 4Beijing Engineering Research Center of Semiconductor Micro-Nano Integrated Technology, Beijing 100083, China

**Keywords:** miniature, piezoresistive pressure sensor, blood pressure, temporary bonding technology

## Abstract

A miniature piezoresistive pressure sensor fabricated by temporary bonding technology was reported in this paper. The sensing membrane was formed on the device layer of an SOI (Silicon-On-Insulator) wafer, which was bonded to borosilicate glass (Borofloat 33, BF33) wafer for supporting before releasing with Cu-Cu bonding after boron doping and electrode patterning. The handle layer was bonded to another BF33 wafer after thinning and etching. Finally, the substrate BF33 wafer was thinned by chemical mechanical polishing (CMP) to reduce the total device thickness. The copper temporary bonding layer was removed by acid solution after dicing to release the sensing membrane. The chip area of the fabricated pressure sensor was of 1600 μm × 650 μm × 104 μm, and the size of a sensing membrane was of 100 μm × 100 μm × 2 μm. A higher sensitivity of 36 μV/(V∙kPa) in the range of 0–180 kPa was obtained. By further reducing the width, the fabricated miniature pressure sensor could be easily mounted in a medical catheter for the blood pressure measurement.

## 1. Introduction

Due to lower power consumption, higher integration level and the ability for small point/space limited measurement, miniature pressure sensors are widely used in industrial/aerospace/medical/consumer electronics fields. For example, these small pressure sensors on the skin of the airfoil can obtain the speed and angle to control flight altitude in the aircraft systems. In the petrochemical/biopharmaceutical fields, multipoint pressure testing is demanded to control the pressure accurately for better quality of the products. For tire pressure monitoring systems (TPMS), miniature pressure sensors can detect the pressure to avoid the explosion and loss of air pressure in tires [1]. For medical diagnosis, these miniature transducers can directly measure the blood pressure in the narrow blood vessels, such as cardiovascular with the inner diameter from 200–1000 μm [2].

According to the working principle, pressure sensors could be divided into piezoresistive, capacitive and resonant sensors. Piezoresistive pressure sensors as an attractive research hotspot have the advantages over simple structure and easy mass production, but suffering from the problems of large size and low sensitivity in comparison to other sensors [3,4,5].

In order to achieve high sensitivity and small size for the piezoresistive pressure sensors, there have been lots of studies focused on the design, simulation and the manufacturing technology have been performed [6,7]. For example, Kalvesten, E. et al. [8] fabricated a thin-film pressure sensor by releasing thermal oxygen sacrificial layer in a cavity with a gap of 3 μm. Gowrishetty U. et al. [9] proposed a miniature pressure sensor using silicon on insulator (SOI) and deep reactive ion etching (DRIE) technologies. The pressure sensor was thinned to 150 μm without back sealing bonding. Li, C. et al. [10], Zhang, B.R. et al. [11], Zou, H. et al. [12] and Meng, X. et al. [13] improved the sensitivity and linearity by designing the island/beam structure on a flat diaphragm. Pramanik, C. et al. [14] and Yu, H. et al. [15] optimized the design of the flat pressure sensor by adjusting the doping concentration of piezoresistive material and the thickness of the diaphragm. In addition, for realizing the miniaturization, reducing the thickness of the sensor and protecting the device from fragmentation in the followed processes are the key steps. When the thickness of a wafer decreases continuously, the subsequent process may normally result in a high probability of chip fragmentation due to the reduction of structural stiffness, thus leading to a low yield.

In this paper, a miniature piezoresistive pressure sensor fabrication process based on temporary bonding technology was proposed, which ensures that the thickness of the device can be as small as 100 μm without the fragmentation. The processes included temporary bonding, chemical mechanical polishing (CMP) and DRIE. The optimum piezoresistive placement, doping concentration and the size of sensing diaphragm were determined by simulation and theoretical calculation. Finally, a half-bridge pressure sensor was designed and fabricated.

## 2. Materials and Methods

When some materials are subjected to small strains caused by external forces, the electronic energy level states of their internal atomic structures will change, resulting in drastic changes in their resistivity, as varied with the crystal directions. This physical effect is called the piezoresistive effect [16]. In this paper, single crystal silicon is used as a pressure-sensitive material because of its high piezoresistive effect. In order to get higher sensitivity, the piezoresistors should be placed at positions with the maximal stress on the diaphragm. Compared with circular and rectangular diaphragms, the square membrane has higher stress distribution and better area utilization [17]. Furthermore, the high thickness-to-length ratio and the noise caused by Brownian motion should be considered, especially when the side length is reduced to 50 μm. Therefore, the square membrane with a size of 100 μm × 100 μm is adopted as the pressure-sensitive diaphragm. In the following sections, the pressure sensor with better performances have been presented based on the designed sensor model.

### 2.1. Thin-Diaphragm Design

The stress distribution of a square membrane in small deflection is studied using Timoshenko’s thin plate bending theory [18,19], and the formula for the deformation of the square diaphragm supported by surrounding force is illustrated as follows:(1)$$w\;=\frac{0.0159p\left(1-u^{2}\right)A^{4}}{EH^{4}}.$$

In the Equation (1), p, u, E, A, H, w are respectively the mean pressure, Poisson’s ratio, Young’s modulus, side length, thickness and ratio of maximum normal displacement to thickness. Based on the small deflection theories of plates [20], the performance will illustrate a good linearity when w ≤ 3% for a dynamic pressure range of 0–180 kPa, so the diaphragm thickness is optimized to 2 μm.

### 2.2. Piezoresistor Design

The pressure sensor is provided with a sensitive piezoresistor on the membrane above a vacuum-sealed chamber and an external reference resistor for temperature compensation. When the piezo resistors are configured with other two resistors as a Wheatstone full bridge, the output voltage proportional to the pressure will be given:(2)$$V_{out}=\left(\frac{R_{3}}{R_{3}+R_{4}}-\frac{R_{2}}{R_{1}+R_{2}}\right)V_{in}.$$

Assuming *R*_1_ = *R*_2_ = *R*_3_ = *R*_4_ = *R*, using the Taylor series to expand Equation (2), the output voltage can be approximately expressed as:(3)$$V_{out}=V_{offset}+\frac{V_{in}}{4}\left(\frac{\Delta R}{R}\right)\approx \frac{V_{in}}{4}\left(\frac{\Delta R}{R}\right).$$

Here $V_{offset}$ is the output voltage at zero stress. In the orthogonal coordinate system, when the axis is consistent with the crystallographic axis, the relative change of resistance in Δ*R*/*R* relationship with stress meets as [21]:(4)$$\frac{\Delta R}{R}=\pi_{l}\sigma_{l}+\pi_{t}\sigma_{t},$$
where $\pi_{l}$ is the longitudinal piezoresistive coefficient and $\pi_{t}$ is the transverse piezoresistive coefficient; $\sigma_{l}$ represents longitudinal stress and $\sigma_{t}$ represents transverse stress.
(5)$$\pi_{l}=\frac{\pi_{11}+\pi_{12}+\pi_{44}}{2}\approx \frac{\pi_{44}}{2}$$
(6)$$\pi_{t}=\frac{\pi_{11}+\pi_{12}-\pi_{44}}{2}\approx -\frac{\pi_{44}}{2}.$$

By substituting Formulas (4)–(6) into Formula (3), the expression of sensitivity is obtained as follows:(7)$$S=\frac{\Delta V_{out}}{pV_{in}}=\frac{\pi_{44}}{8p}\left(\sigma_{l}-\sigma_{t}\right).$$

According to this formula, the shear piezoresistive coefficient and the difference between longitudinal stress and transverse stress are the main factors affecting the sensitivity *S*, determined by the design and placement of the piezoresistors. 

A half-bridge structure with three aluminum electrodes arranged in transverse order has been designed in order to reduce the area of the sensor, as shown in Figure 1. Form the section along with A_1_-A_2_, a “buried” piezoresistor embedded in the diaphragm for the circuit, is electrically isolated from the silicon through the PN junction. The other is located outside the diaphragm as a reference resistor. There is an insulating layer of silicon oxide between the top silicon and the metal interconnection, which is not shown in the figure. The piezoresistor is connected to the metal interconnection outside the membrane through contact holes of the ohmic contact area.

The stress distribution of silicon thin films is analyzed in detail with the finite element analysis, and the result is shown in Figure 2a. Along the centerline of the diaphragm, the tensile stress at the center of the membrane edge reaches its maximum. Then after a parabolic trend, another maximum value emerges in the center of the diaphragm, which is the compressive stress. However, as can be seen from the stress curve in Figure 2b, the orthogonal stress difference at the center is zero. Therefore, the piezoresistor should be placed at the center of the edge of the diaphragm.

The p-type piezoresistor with “two legs” is produced by boron implantation into the n-type (100) substrate, which has a higher piezoresistive coefficient than the n-type. When the doping concentration is between 3–8 × 10^18^ cm^−3^, the temperature coefficient of sensitivity is relatively small. With the injection energy of 30 keV and an injection dose of 1.7 × 10^14^ cm^−2^, the desired concentration of 7 × 10^18^ cm^−3^ is obtained, after annealing at 1050 °C for 10 min. The power consumption of the bridge is usually tens of milliwatts, which satisfies the requirement of maximum power consumption per unit surface area for monocrystalline silicon, expressed in Equation (8).
(8)$$p=\frac{I^{2}R}{WL}=\frac{I^{2}\rho a\frac{L}{W}}{WL}=\frac{I^{2}\rho a}{W^{2}}.$$

In Equation (8), *I*, *ρ*, a, W are respectively the current flowing through the resistors, the resistivity, the cross-sectional area, and the width of the resistors. In consideration of lithographic errors and large stress distribution, the size of 5 μm × 12 μm (*W* × *L*) is adopted for one leg with a sheet resistance of 400–450 Ω and the distance between two legs is of 5 μm.

## 3. Pressure Sensor Fabrication

The fabrication process of the pressure sensor includes temporary bonding and anodic bonding, which make the device into a sandwich structure and protect the wafer in the subsequent processing. All steps are given below, which are briefly described in the caption of Figure 3.
(a)The production begins with an n-type (100) SOI wafer with the thickness of 405 +/−5 μm, on which is the device layer with the thickness and resistivity of 2 μm and 1–10 Ω/cm. The thickness of the buried oxide layer of the SOI wafer is of 0.4 μm.(b)Twice ion implantation on the device layer is performed to form the gauge piezoresistors and ohmic contact area respectively.(c)Silicon oxide as the insulating layer is deposited and the holes in the low resistance region are patterned and etched, followed by aluminum deposition as interconnection and electrodes with a thickness of 300 nm.(d)Sputtering deposition of Chromium/Copper (50 nm/500 nm) as the temporary bonding material and lifting-off to expose the effective device area.(e)Repeating the previous step (d) on a piece of BF33 wafer. The thickness of the BF33 is about 520 μm, and the surface roughness is less than 5 nm. It has high chemical stability and excellent mechanical flexibility, which can be used for wafer carriers bonded to silicon anodes and thin silicon wafers.(f)Bonding the device layer and the BF33 together through the thermocompression bonding technology.(g)Thinning handle layer with CMP (Chemical Mechanical Polishing) to a thickness of 60 μm and etching the handle layer after back-aligned lithography with silicon oxide as the etch stop layer to fabricate the cavity.(h)Sputtering a layer of aluminum on the upper BF33 by magnetron sputtering technology. The edge of the structure is also covered with aluminum, so that the upper layer of aluminum is electrically connected to the lower SOI sheet for the next anode bonding. Then, bonding the structure to another BF33 wafer by anodic bonding in a vacuum chamber at 5 × 10^−4^ Torr and polishing the BF33 to a thickness of 40 μm.(i)Dicing the device into the size of 1600 μm × 650 μm and separating the structure by immersing in a special etchant (H_2_SO_4_:H_2_O_2_:H_2_O = 1:1:40).

Using temporary bonding technology, the first BF33 can play a positive role in the CMP process. At the same time, the stiffness of the whole structure is guaranteed to avoid the fragmentation, caused by excessively the thin wafers in subsequent processes. With the special etchant, it is easy to separate the structure and does not corrode aluminum distinctly. Finally, we have made a miniature pressure sensor that has a size of 1600 μm × 650 μm, as shown in Figure 4. From the illustration at the top right, the remaining thickness of the SOI and BF33 wafer is 77 μm and 27 μm. Therefore, the overall thickness of the chip is about 104 μm.

## 4. Results and Discussion

In order to verify the performance of the fabricated pressure sensor, the diced sensor chip is assembled in a Kovar base with the adhesive plaster, as shown in Figure 5a. The Kovar base is then mounted into the metal envelope using a sealing rubber ring. The pins are connected to the automatic stress test system via signal lines. The pressure was applied by using a pressure controller (GE Pace6000, The General Electric Company, Boston, MA, USA), and the 1 mA input of constant current was provided by a digital source-meter (KEITHLEY 2450, Tektronix, Cleveland, OH, USA) to provide a constant current source of 1 mA for alleviating the influence of the temperature on the sensor. The output voltage signal has been detected by using a multimeter (HP 34401A, Agilent Technologies Inc., Santa Clara, CA, USA). These three devices are connected and controlled through a program, set by LabVIEW. When the pressure range and settling time are already set, the output voltage signal can be measured automatically. The pressure test system is shown in Figure 5b.

The *I–V* tests has been performed with a Semiconductor Device Parameter Analyzer (Keysight B1500, Aglient, Palo Alto, CA, USA). The resistance value of piezoresistors and reference resistors are between 1.8–2.2 kΩ, which are very close to the design value of 2.1 kΩ. When a constant current source of 1mA is applied, multiple cycle tests on the pressure sensor have been carried out. In the pressure range of 0–180 kPa, a sensitivity of 36 μV/(V∙kPa) is obtained at room temperature. Figure 6 shows three sets of data for circular journey at 25 °C.

To characterize the output performances with temperature, the pressure sensors are tested by mounting them. As shown in Figure 7a, the sensitivity is still maintained at around 36 μV/(V∙kPa) in the temperature range of the human body (34–42 °C). In the temperature range of 25–95 °C, although the chip still exhibits good linearity, the sensitivity gradually decreases with the increase of temperature, as illustrated in Figure 7b.

At the same time, from Figure 8, the zero output drifts at different temperatures. The reason for this result is that, as temperature increases, the gas generated during anodic bonding will expand, causing a change in the resistance on the membrane. In addition, due to the half-bridge design without the temperature compensation circuit, the temperature drift will be relatively obvious. Table 1 lists the performance of the miniature pressure sensor. The test results show that the chip has acceptable performance in terms of sensitivity, linearity, repeatability, and hysteresis. In addition, in the temperature range of 34 to 42 °C, the temperature coefficient of sensitivity (TCS) and the temperature coefficient of zero offset (TCO) are 1.03 × 10^−3^ FSO/°C and 8.11 × 10^−4^ FSO/°C respectively, indicating the good temperature characteristics.

The temporary bonding process provides the rigidity support for thinning pressure sensor, and to a certain extent, avoids the possibility of the thin wafer fragmentation in the subsequent process. The final protective BF33 is easy to separate away, without affecting the effective structural region, and the overall fabrication process is relatively simple. As can be seen from Table 2, compared with the research works that have been reported, the fabricated pressure sensor achieves a reduction in the thickness direction, meanwhile ensuring a good performance.

## 5. Conclusions

A miniature piezoresistive pressure sensor is proposed and fabricated with temporary bonding technology. Through finite element simulation and theoretical calculation, the parameters of piezoresistors and the diaphragm structure are optimized. In the process of fabrication, the temporary bonding sheet plays a protective role for the device to achieve a high yield. With the help of CMP and RIE, the small pressure sensor chip is achieved with a full size of 1600 μm × 650 μm × 104 μm. Under the pressure range of 0–180 kPa, a sensitivity of 36 μV/(V∙kPa) is obtained. In the range of human body temperature, the pressure sensor shows excellent linearity of 0.141%. After subsequent optimization of the circuit layout and rearrangement of the cutting lines, dimensions of the chips can be further reduced and thus the fabricated miniature pressure sensors are potentially used in medical blood pressure measurement or aerospace pressure monitoring as well as the other fields due to its small size.

## Figures and Tables

**Figure 1 sensors-20-00337-f001:**
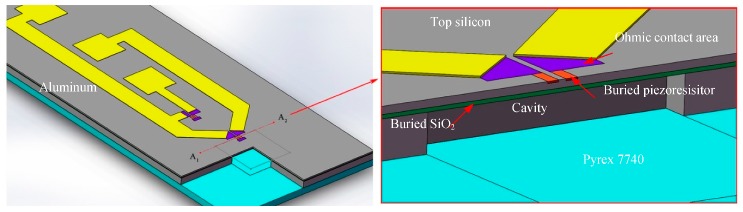
Sketch of the proposed pressure sensor.

**Figure 2 sensors-20-00337-f002:**
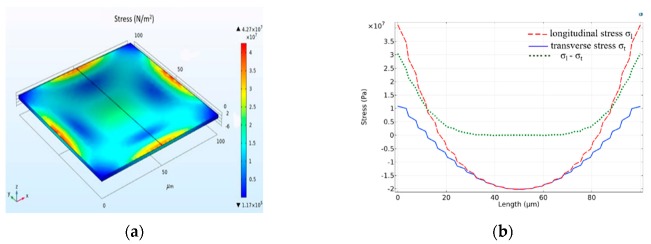
Stress distribution under finite element analysis. (**a**) Stress distribution at 60 kPa; (**b**) longitudinal and transverse stress distribution.

**Figure 3 sensors-20-00337-f003:**
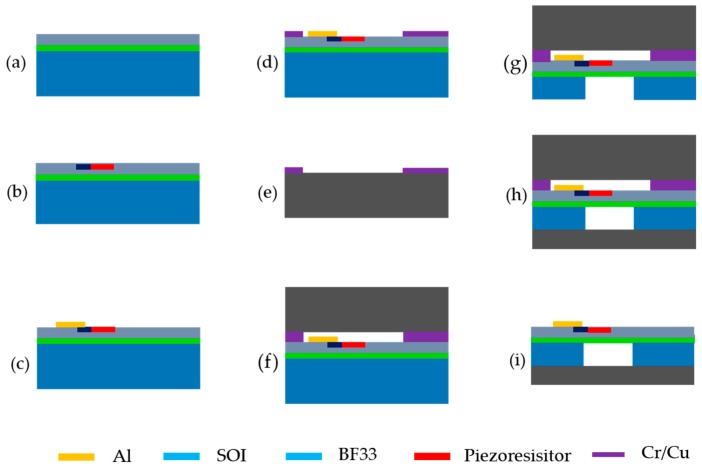
Manufacturing process of the proposed pressure sensor (**a**) SOI wafer, (**b**) ion impantation, (**c**) the electrode preparation, (**d**) sputtering deposition, (**e**) BF33 wafer, (**f**) bonding, (**g**) CMP and erching, (**h**) bonding and polishing, (**i**) dicing and separating.

**Figure 4 sensors-20-00337-f004:**
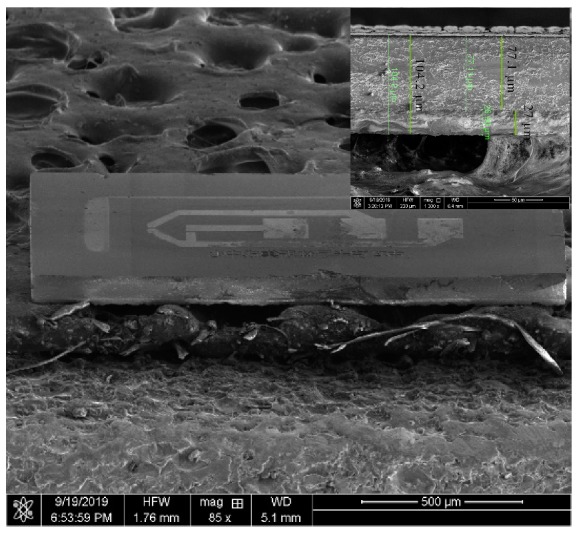
SEM image of the sensor die after separating the structures. The insert shows that there is a visible boundary between BF33 and the handle layer.

**Figure 5 sensors-20-00337-f005:**
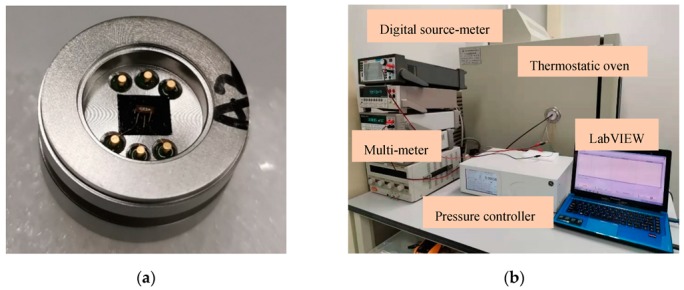
Photographs of calibration equipment. (**a**) Miniature pressure sensor assembled in a Kovar base. (**b**) Automatic pressure test system.

**Figure 6 sensors-20-00337-f006:**
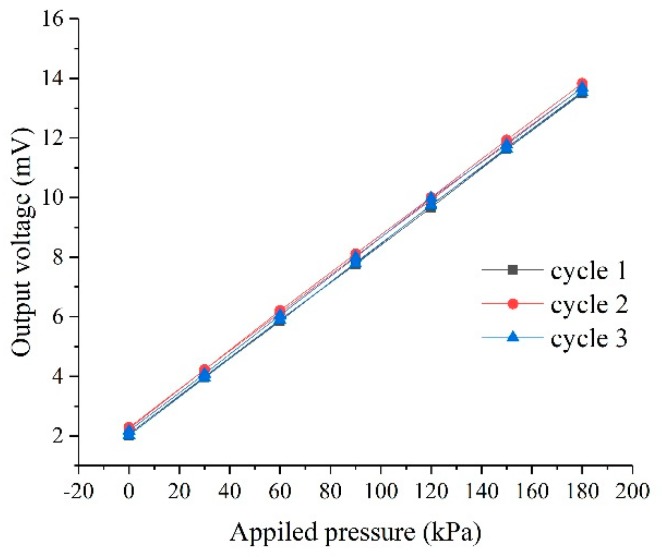
Relationship between output voltage and applied pressure.

**Figure 7 sensors-20-00337-f007:**
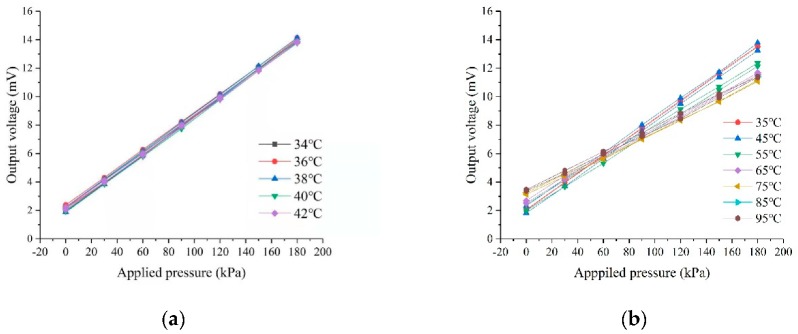
Output voltage for different temperatures. (**a**) for the human body temperature range. (**b**) for temperature range from 25 to 95 °C.

**Figure 8 sensors-20-00337-f008:**
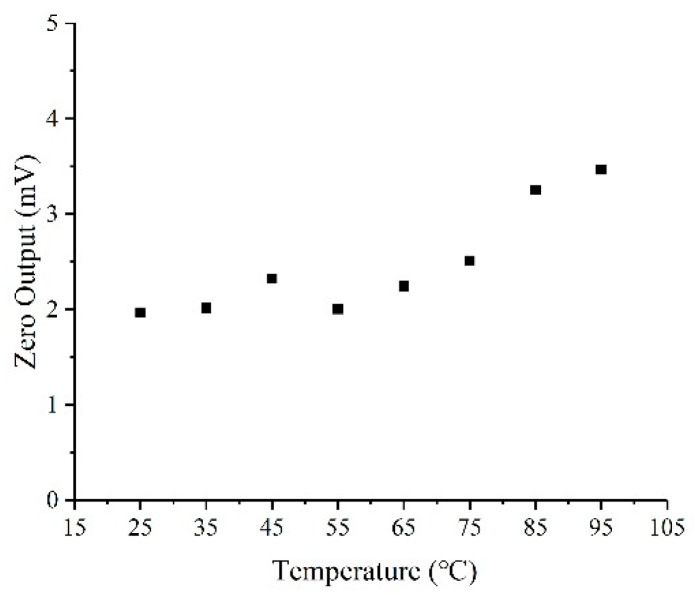
Zero output with different temperatures.

**Table 1 sensors-20-00337-t001:** Specifications of the miniature pressure sensor.

Parameters	Values	Units
Diaphragm size	100 × 100	μm
Chip size	1600 × 650 × 104	μm
Sensitivity	36	μV/(V∙kPa)
Nonlinearity	0.141	% FSO
Hysteresis	0.476	% FSO
TCO (34–42 °C)	8.11 × 10^−4^	FSO/°C
TCS (34–42 °C)	1.03 × 10^−3^	FSO/°C
TCO (25–95 °C)	0.186 × 10^−4^	FSO/°C
TCS (25–95 °C)	4.46 × 10^−3^	FSO/°C

**Table 2 sensors-20-00337-t002:** Comparison with other miniature pressure sensors.

Chip Size (mm)	Sensitivity (μV/(V∙kPa))	Nonlinearity (% FSO)	Reference
1.6 × 1.6 × 0.9	27.9	0.34	[22]
1.25 × 1.25 × 0.45	30	-	[23]
1.0 × 1.0 × wafer thickness	52.2	0.1	[24]
1.3 × 1.3 × 0.55	28.9	0.21	[25]
1.6 × 0.65 × 0.1	36	0.141	Fabricated sensor

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
