# Peer review of "A Novel Piezoresistive MEMS Pressure Sensors Based on Temporary Bonding Technology"

_sensors, 2020, doi:10.3390/s20020337_

Round 1
Reviewer 1 Report
1. The manuscript must be carefully proof-read to improve English and correct a number of grammar mistakes.
2. From the title and abstract, it seems that the focus of this paper is the temporary bonding technology or fabrication technology. But actually it is not. So the title and abstract need to be revised. For instance, the title can be: Piezoresistive MEMS Pressure Sensors Based on a Temporary Bonding Technology
3. A reference should be given to Equations (4) and (5). Also note that they are true for p-type silicon. ]
4. Lines 107-110: it is better to call “the film” diaphragm or membrane. Please make this change for other places as well.
5. Step (h) needs to be explained in more details. Now the top surface of the silicon wafer is bonded with another glass wafer. How does the electrode for anodic bonding make contact with the silicon wafer?
6. Please give a brief introduction of BF33 wafer.
7. The fonts on insert in Fig. 3 are too small.
8. Please briefly discuss potential contamination of Cu on silicon (Cu is a fast diffuser in silicon)
Author Response
Thanks for your careful reading of our manuscript. It’s very valuable and helpful for revising and improving our paper.
Please see the attachment.

Reviewer 2 Report
Overall, sound scientific work and well-written paper.
Please consider the following corrections or comments:
The term “membrane” is not correct for your pressure sensor. What you use is a “bending plate” because the bending stiffness of the plate is relevant, not the tension of a membrane In all graphs the numbers and letters are too small. Line 62: replace “was” by “is”. Line 101: You should use a lower case “P” in Formula (7) like in Formula (1). Figure 2: In Figure 2b it is unclear which line shows which data. Better use “solid line”, “dashed line” and “dotted line”. Line 151, Figure 3: On my printout of the paper the different layers can not be seen. Please increase the size of the pictures and the thickness of the layers.
I would prefer if blue is used for Si and grey for the BF33 Glass. That is the standard colour scheme. Line 162: Please add the Information that this is a vacuum bonding process. Line 178, Figure 3: Use new number for Figure. The brightness and the contrast of the SEM image are too low. Line 181: Replace “Discuss” by “Discussion”. Line 189: Replace “output” by ”measured”. Figure 4a: Please Increase the size of the interesting inner part of the picture. Line 205: Please replace caption by: “Output voltage for different temperatures. (a) for human body temperature range. (b) for temperature range from 25 °C to 95 °C. “. Figure 7 and Line 209: What do you mean by “Zero output”? The output voltage at a pressure of 0 kPa? (Than the values are not in agreement with the values shown in Figure 6(b) ). Line 209: No correct English sentence. Line 210: I assume that the bonding process is in a vacuum chamber so there should be no gas in the cavity. Line 218, Table 1: Please define how the hysteresis was calculated. Is it a Pressure-Hysteresis or a Temperature-Hysteresis? Which values and times were used for the measurement?
Author Response
Thanks for your good instruction. It’s very valuable and helpful for revising and improving our paper.
Please see the attachment.

Reviewer 3 Report
Song and co-authors introduced a novel manufacture method for development of piezoresistive MEMS pressure sensor based on temporary bonding technology. The size of a sensing membrane is 100 μm × 100 μm × 2 μm, the small pressure sensor is 25 obtained with a full size of 1600 μm × 650 μm × 104 μm, and the pressure sensor has a high sensitivity 26 of 36 μV/V/kPa in the range of 0–180 kPa. These sensing devices are promising for application in medical diagnosis. Paper can be published on Sensor after minor revision.
As the authors mention this is novel manufacture method, therefore authors should present the advantage of this method in comparison with other published methods to fabricate pressure sensor. A table comparing the performance of this pressure sensor with the state of the art published elsewhere (other piezoresistive pressure sensor) would be helpful. Response time and relaxation time of sensing device under applied dynamic pressure should be provided.
Author Response

(The authors gave the same response as above.)
